# Flora and Vegetation of Yunnan, Southwestern China: Diversity, Origin and Evolution

Hua Zhu * and Yunhong Tan

Center for Integrative Conservation, Xishuangbanna Tropical Botanical Garden, Chinese Academy of Sciences, Xishuangbanna 666303, China; tyh@xtbg.org.cn
* Correspondence: zhuh@xtbg.ac.cn

**Abstract:** Yunnan has a complicated geological history, a particular geography, and a complex topography, which have influenced the formation of various habitats of high biodiversity: 245 families; 2140 genera; 13,253 species and varieties of seed plants; more than 12 types of vegetation; and 167 plant formations, including tropical rain forests, tropical dry forests, subtropical evergreen broad-leaved forest, cold temperate coniferous forests, and alpine bushes and meadows. An analysis of the geographic elements to the current Yunnan flora shows that the tropical distribution contributed to 51% of all families and to 57.5% of all genera, of which the genera from the tropical Asian distribution make up the highest proportion among all geographical elements. During the late evolution of Yunnan, its flora was strongly affected by the tropical Asian flora. The complicated patterns and diversity in Yunnan flora and vegetation have been shaped mainly by its particular geological histories, which include the differential uplifts in topography, the clock-wise rotation of the Simao-Lanping geoblock, and the extrusion of the Indochina geoblock by the Himalayan uplift. The flora and vegetation of Yunnan were possibly derived from tropical-subtropical Tertiary flora before later diverging. Northwestern Yunnan flora likely evolved due to rapid speciation from families and genera from cosmopolitan and northern temperate distributions during the uplift of the Himalayas and climatic oscillations after the late Tertiary. Southern Yunnan flora likely evolved into tropical Asian flora following the southeastward extrusion of the Indochina block, which brought along tropical Asian elements. Central Yunnan flora inherited most of the elements of the Tertiary flora from East Asia. The formation and strengthening of the southwest monsoon by the uplift of the Himalayas was also a direct factor in the formation of the tropical rain forests found in southern Yunnan. The flora from southern and southeastern Yunnan also diverged, with the former being more closely related to Indo-Malaysian flora and the latter being more closely related to Eastern Asian flora. This floristic divergence is well supported by the geological history of these regions: that is, the tropical flora of southeastern Yunnan derived from the South China geoblock, whereas the flora of southern and southwestern Yunnan mainly derived from the Shan-Thai geoblock.

**Keywords:** flora; vegetation; origin; evolution; Yunnan; SW China

## 1. Introduction

Yunnan is biogeographically located at the junction where tropical Asia, East Asia, and the Himalayan subtropical-temperate zones meet (Figure 1). With a complex topography consisting of deep valleys, plateaus, and mountains, the change in elevation is huge (from the 76 m altitude at the southeastern-most boundary to the 6740 m altitude at Kavagbo Peak of the Meili Mountains in the northwestern part), decreasing from north to south (Figure 2). Geologically, it is a suture zone between Gondwana and Laurasia and its territory mainly formed when the Himalayas rose during the Tertiary, parting from the ancient south China geoblock. The Gondwana and Laurasia flora contributed to the evolution of the flora, and the current flora shows a mixture of elements from tropical Southeast Asia, East Asia, and the temperate Himalayas. Yunnan was also affected by the Himalayan uplift and the

accompanying geological events and large river captures. Due to its exceptional geological, geographical and topographical features, Yunnan has evolved to have high biodiversity.

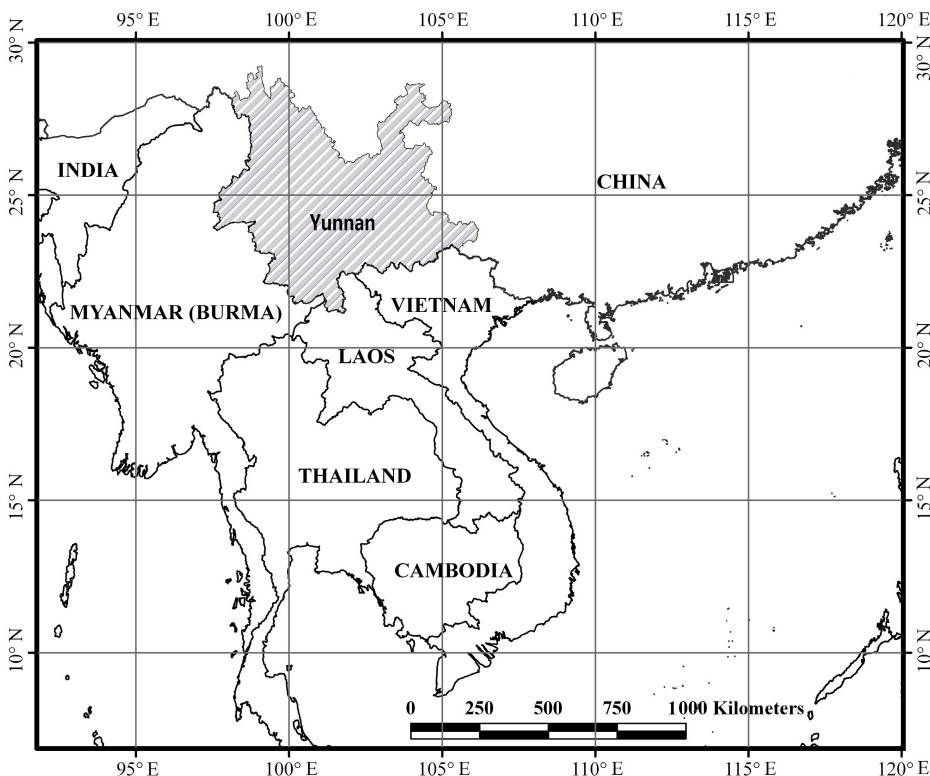

**Figure 1.** Geographical location of Yunnan (this figure was made by the Landscape Ecology Lab., Xishuangbanna Tropical Botanical Garden, CAS).

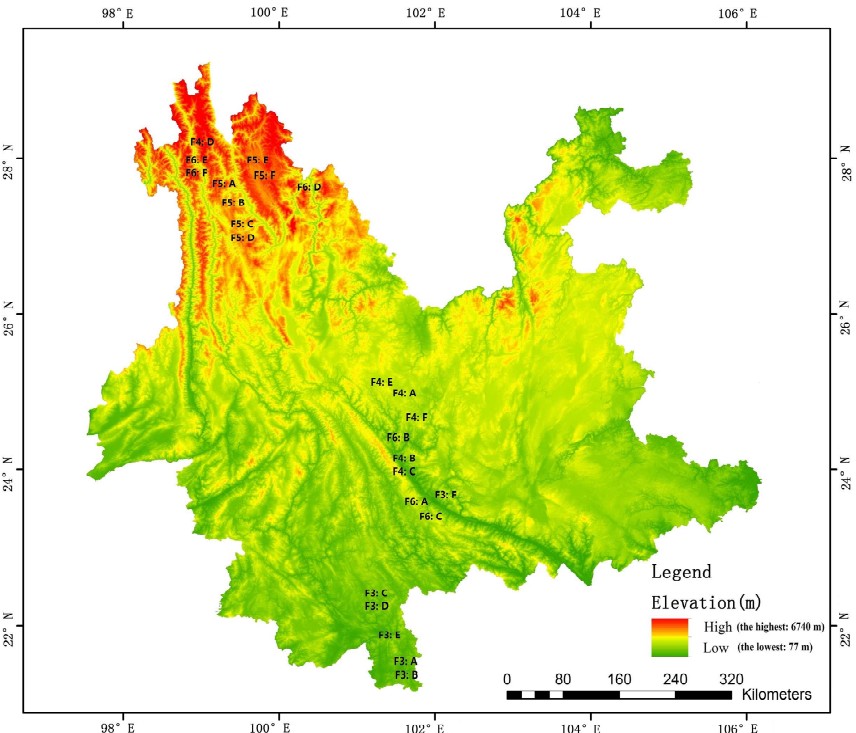

**Figure 2.** Topography of Yunnan with photo sites taken for vegetation types (this figure was made by the Landscape Ecology Lab., Xishuangbanna Tropical Botanical Garden, CAS; F3, F4, F5, F6 in the map are the shortening of legends in Figures 3–6).

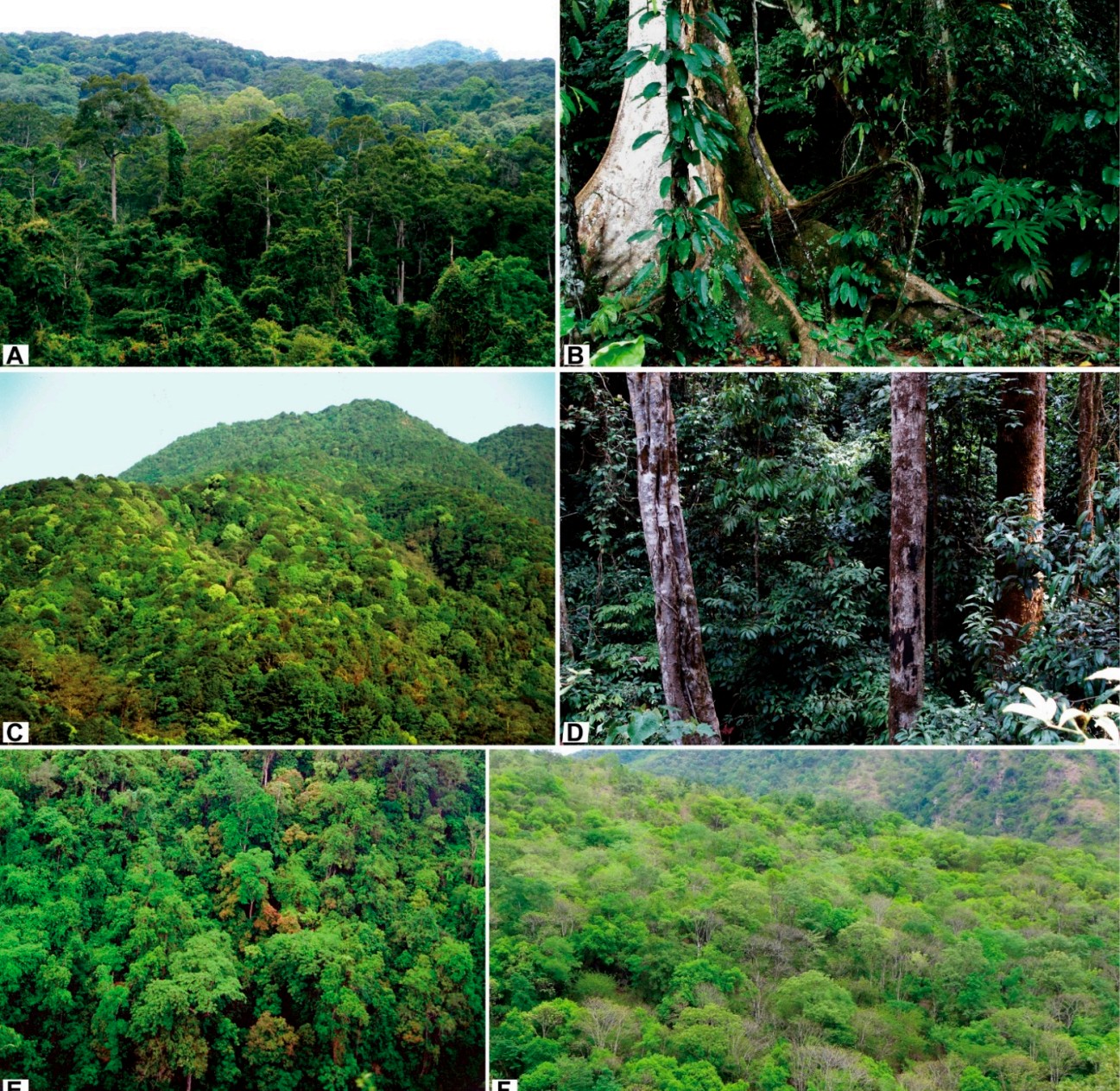

**Figure 3.** (**A**), Tropical rain forest in southern Yunnan; (**B**), understory of the tropical rain forest in southern Yunnan; (**C**), tropical lower montane evergreen broad-leaved forest in southern Yunnan; (**D**), understory of the tropical lower montane evergreen broad-leaved forest in southern Yunnan; (**E**), tropical seasonal moist forest on limestone in southern Yunnan; (**F**), tropical deciduous forest in southern Yunnan.

Yunnan takes up only 4.1% of the total land area of China but includes more than half of the total plant species found and has various types of vegetation, from tropical to cold temperate forests. Based on an analysis of herbarium specimens collected, 245 families, 2140 genera, and 13,253 species and varieties of seed plants were confirmed to be found in Yunnan [1]. Additionally, from the data recorded from field surveys, the main types of vegetation found in Asia are also found in Yunnan [2]. For example, forests similar in floristic composition and physiognomy to the tropical rain forests of SE Asia were found in southwestern, southern, and southeastern Yunnan [3–16]; tropical deciduous forests and savanna-like vegetation found in hot dry valleys [17,18], and the subtropical

evergreen broad-leaved forests of eastern Asia were found also in the central plateau of Yunnan [19–21]; cold temperate coniferous forests dominated by spruce, fir, and larch trees and temperate deciduous forests dominated by birch wood, maple, and poplar trees were found in high northern mountains; and sclerophyllous evergreen broad-leaved forests dominated by the *Quercus* species were commonly found in diverse habitats throughout Yunnan, from hot, dry, and deep valleys to cold temperate mountains. The monograph "Vegetation of Yunnan" by Wu recorded 12 main natural types of vegetation [22], which includes all of the main types of vegetation found in China.

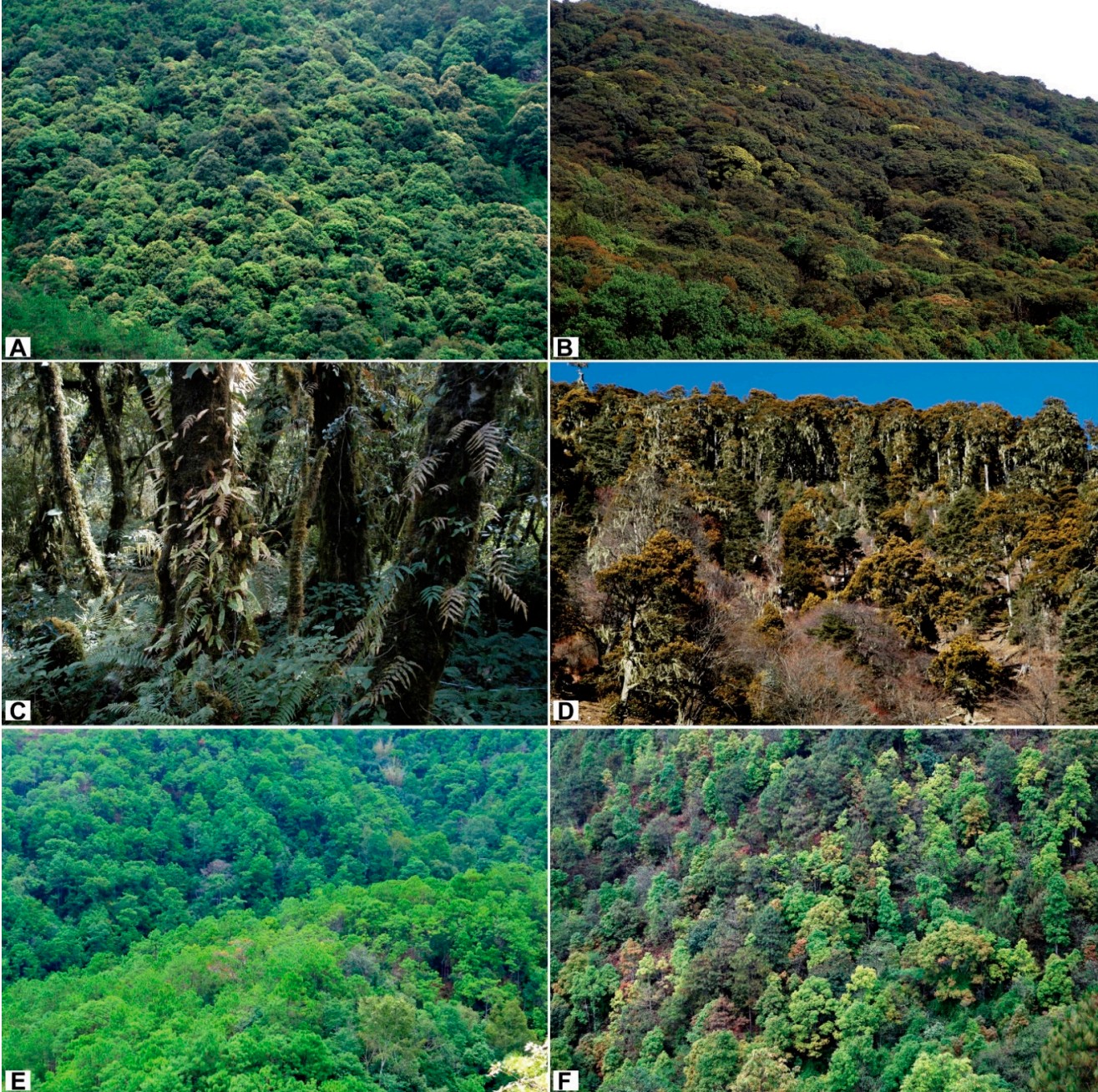

**Figure 4.** (**A**), Semi-evergreen broad-leaved forest in central Yunnan; (**B**), mid-montane wet evergreen broad-leaved forest in central Yunnan; (**C**), understory of the mid-montane wet evergreen broad-leaved forest, showing abundant epiphytes; (**D**), sclerophyllous oak forest in northern Yunnan; (**E**), temperate coniferous forest in central Yunnan; (**F**), temperate coniferous and evergreen broad-leaved mixed forest in central Yunnan.

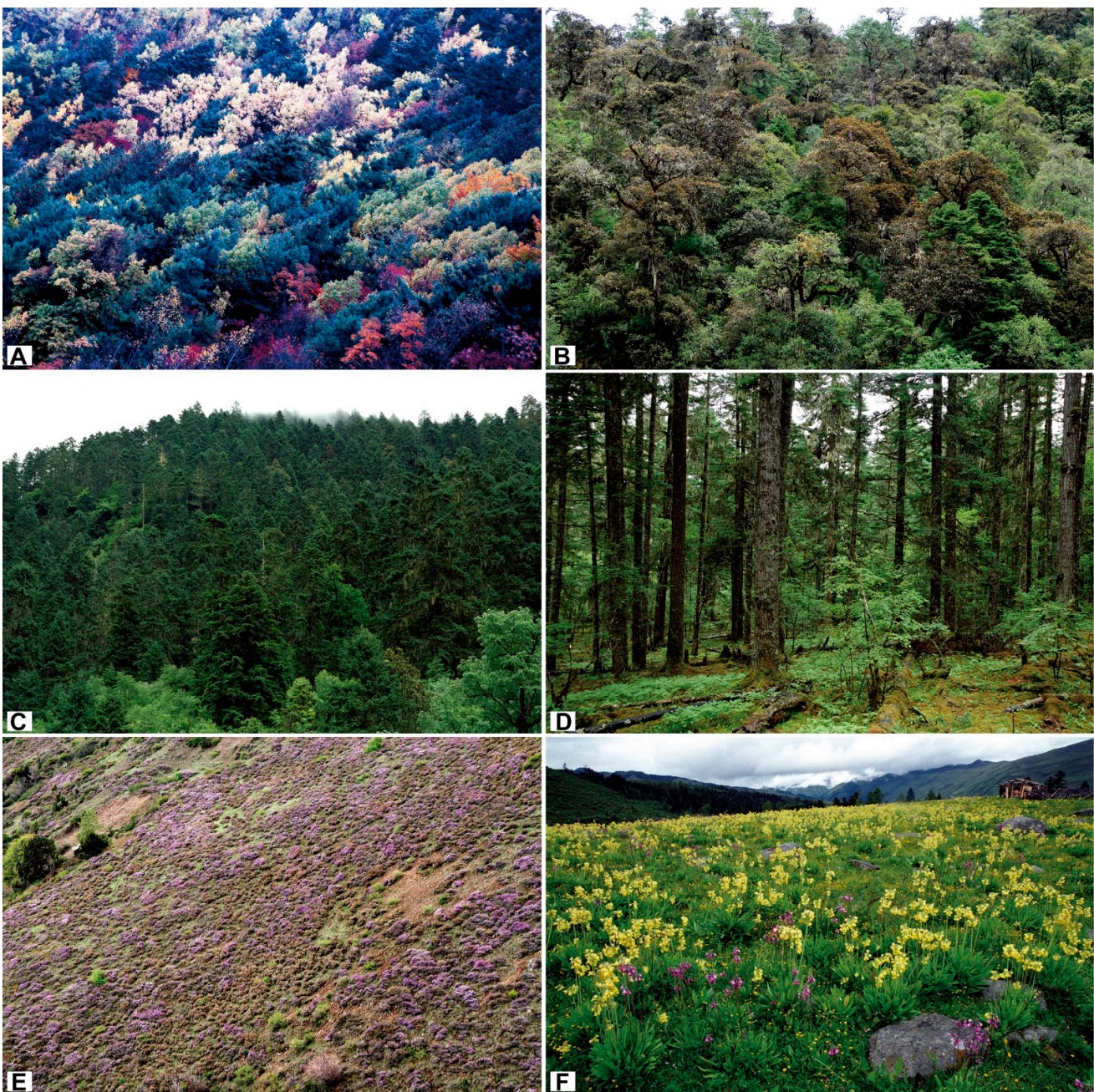

**Figure 5.** (**A**), Temperate coniferous and deciduous broad-leaved mixed forest in northern Yunnan; (**B**), cold temperate coniferous and sclerophyllous oak mixed forest in northern Yunnan; (**C**), cold temperate coniferous forest in northern Yunnan; (**D**), understory of the cold temperate coniferous forest in northern Yunnan; (**E**), alpine Rhododendron shrubs in northern Yunnan; (**F**), alpine meadows in northern Yunnan.

Geological events that have occurred since the Tertiary period—such as the differential uplift of Yunnan; the clock-wise rotation of the Simao-Lanping geoblock; the southeastward extrusion of the Indochina geoblock caused by the collision of India with Asia; and the strengthening of the southwestern Asian monsoon climate triggered by an uplift in the Himalayas as well as several large rivers that flow southward, which were once tributaries to a single, southward flowing system (the paleo-Red River since at least the late-Miocene period [23])—influenced the origin and evolution of Yunnan flora and vegetation [1,9–11,17,24–29]. These geological events also deeply influenced the divergence

of flora in Yunnan, such as the formation of some suggested biogeographical demarcation lines [24,26,30–34].

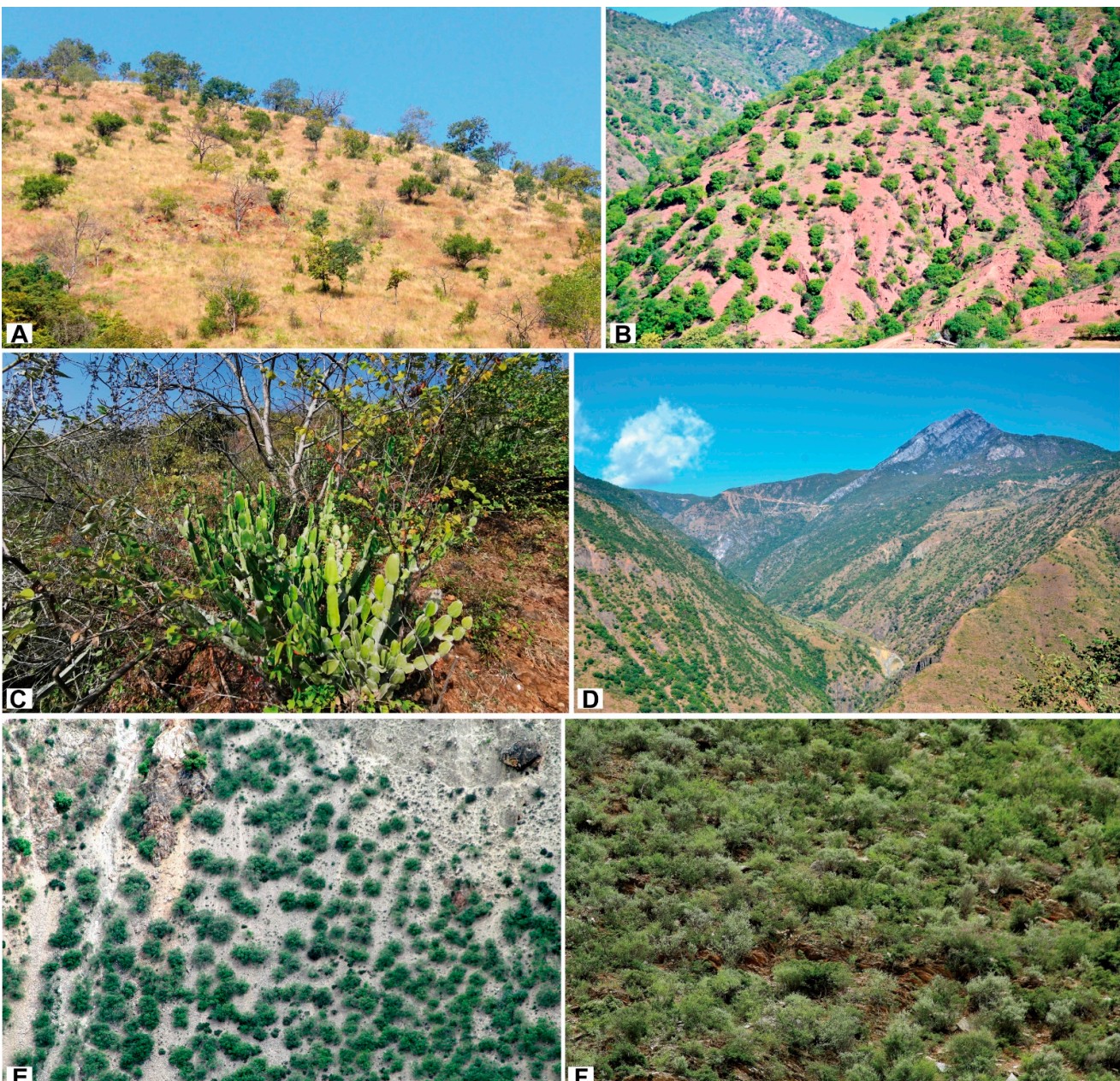

**Figure 6.** (**A**), Savanna-like vegetation in hot dry valleys in southeastern Yunnan; (**B**), savanna-like shrubs in hot dry valleys in southeastern Yunnan; (**C**), succulent shrubs in hot dry valleys in southeastern Yunnan; (**D**), savanna-like vegetation in hot dry valleys in northern Yunnan; (**E**), maquis-like vegetation in warm dry valleys in northern Yunnan; (**F**), close up of the maquis-like vegetation in warm dry valleys in northern Yunnan.

However, the origin and evolution of the flora and vegetation of Yunnan are still somewhat unclear. Paleobotanical studies offer good clues about their origin and evolution. The divergence of the evergreen broad-leaved forests of Yunnan was suggested to have started in the late Miocene [35–37]. During the Miocene, the vegetation of southwestern Yunnan was suggested to be tropical in nature, while that of eastern and northern Yunnan were suggested to be subtropical [38]. The preliminary composition and distribution of the present flora and vegetation of Yunnan formed during the Miocene [27,37]. Based on

paleobotanical and molecular phylogeny studies of the Hengduan Mountains, psychro-phytes grew during the early Oligocene [39]; some current genera and species, such as *Quercus, Alnus, Betula, Carpinus, Carya, Pterocaryae*, etc., appeared in the Lühe basin of central Yunnan in the early Oligocene [40]; and subtropical forests supposedly formed in the central Qinghai-Tibetan Plateau 47 million years ago (Ma) [41]. Paleobotanical data can serve as evidence of the origin and evolution of the flora and vegetation of Yunnan.

Most studies on the flora of Yunnan focus on local flora in nature reserves, while research of the regional and provincial floras is rare: only found in the studies by Li, Li and Walker, and Zhu [1,42,43] and in the monograph by Wu [44]. Although some studies on the origin and evolution of Yunnan have been published [11,27,45,46], synthetic studies combining its paleobotanical, geological, and climatic histories are still needed.

A descriptive work on the vegetation in Yunnan was first published by Wang [47]. Later, Wu published the monograph *Vegetation of Yunnan* [22], which laid the foundation for studies of the vegetation in Yunnan. Zeng published a study on the natural forests of Yunnan based on vegetation classification and distributions [48]. Zhu discussed the main types of vegetation in Yunnan, and their origin and evolution [2]. However, more work on the extremely diverse vegetation of Yunnan is further needed.

Combining community ecology and floristic plant geography to study types of vegetation could give a better overall understanding of vegetation: not only its ecological and physiognomic features, but also its biogeographical sources. This article aims to review and clarify the diversity, origin, and evolution of the flora and vegetation of Yunnan and to provide information useful for biodiversity conservation.

## 2. Composition and Characteristics of the Flora and Vegetation of Yunnan

### 2.1. Floristic Composition and Geographical Elements

We identified 245 families, 2140 genera, and 13,253 species and varieties of seed plants in Yunnan [1]. The species-rich families included Poaceae (874 species), Asteraceae (787), Orchidaceae (774), Fabaceae (637), Rosaceae (460), Lamiaceae (446), Rubiaceae (365), Ericaceae (360), and more. The families with the highest species richness were cosmopolitan ones. Of the geographical elements, the tropical distribution contributed to 52.7%, and the pantropic distribution obtained the highest ratio, making up 34.4% of all families of flora in Yunnan, such as Acanthaceae, Anacardiaceae, Annonaceae, Apocynaceae, Araceae, Arecaceae, and Clusiaceae. The northern temperate distribution contributed to 12.5% of all families, such as Adoxaceae, Betulaceae, Caprifoliaceae, Cornaceae, Fagaceae and Hamamelidaceae. The flora of Yunnan included 12 families from the tropical Asian distribution, such as Crypteroniaceae, Iteaceae, Ixonanthaceae, Pentaphragmataceae, Pentaphylacaceae, Sabiaceae and Sladeniaceae, and 8 families from the eastern Asian distribution, such as Cercidiphyllaceae, Circaeasteraceae, Eupteleaceae and Stachyuraceae. The families from the tropical distribution were found mainly in southwestern to southeastern Yunnan and in hot dry valleys of several large rivers and their tributaries, while those of a temperate distribution were found mainly on the plateau, and in northwestern and northern Yunnan.

Floristic evolutionary history could be obtained from geographical elements at the generic level in flora. Of the 2140 genera in Yunnan flora, the tropical distribution contributed to 57.4%, of which those of the tropical Asian distribution made up the highest ratio among all geographical elements, contributing 22.2% of all genera. The temperate distribution contributed to 33%, of which the north temperate contributed 10.9%, followed by the eastern Asian distribution, contributing 9.9% of all genera [1]. Tropical elements contributed the most to the flora in Yunnan, showing its tropical affinity.

### 2.2. Vegetation Types and Distribution

From the monograph *Vegetation of Yunnan* [22], 12 vegetation types and 167 formations were identified, including tropical rain forests, subtropical evergreen broad-leaved forests, sclerophyllous oak forests, warm temperate deciduous forests, temperate coniferous and

broad-leaved mixed forests, cold temperate coniferous forests, subalpine and alpine shrubs and meadows, and savanna-like vegetation from hot dry valleys, etc. (see Figures 3–6).

Due to the particular landform and topography of Yunnan, the climate and natural vegetation change dramatically at very short distances. Various types of vegetation display complicated distribution patterns, not only at the provincial but also at the regional levels. Generally, tropical rain forests and tropical deciduous forests are distributed in southwestern, southern, and southeastern Yunnan; subtropical evergreen broad-leaved forests, coniferous and broad-leaved mixed forests, and warm temperate deciduous forests are mainly distributed on the central plateau; and cold temperate coniferous forests are distributed in northern Yunnan. Savanna-like vegetation can be found in the hot dry valleys of several large rivers, and sclerophyllous and microphyllous shrubs and maquis-like vegetation with a Mediterranean floristic affinity can be found in the warm dry valleys in northern Yunnan. Sclerophyllous oak forests dominated by *Quercus franchetii* and *Q. cocciferoides* are found mainly in the hot dry valleys, while those dominated by *Q. guajavifolia* and *Q. semecarpifolia* are found mainly in the central plateaus and in the high mountains of northwestern Yunnan.

Some arguments have been made about vegetation zonation and vegetation regionalization in Yunnan. Liu et al. argued that the vegetation distribution in Yunnan should be regarded as a large vertical distribution based on altitude and that areas below 1500 m in altitude should be considered tropical regions because the topography of Yunnan is generally sloped to the south and most of the climate in Yunnan is affected by the Indian Ocean [49]. Ren and Xiang also suggested that Yunnan is a big tropical mountain with clearly vertical zonality [50]. We support these points of view on vegetation and flora in Yunnan because tropical climates and tropical vegetation occur below 1300 m in altitude in Yunnan regardless of the latitude [51]. However, the vegetation distribution in Yunnan can also be regarded based on latitude [52,53], as was accepted in the monograph *Vegetation of Yunnan* [22].

*2.3. Biogeographical Divergence of the Flora and Vegetation of Yunnan*

On floristic regionalization, Wu and Wu raised the "Eastern Asiatic floristic region" delineated by Takhtajan [54] to the kingdom level, "the Eastern Asiatic Kingdom", and further divided it into the Sino-Himalaya and Sino-Japanese Subkingdoms [55]. The demarcation line between the Sino-Himalaya and Sino-Japanese Subkingdoms runs north–southward in eastern Yunnan. This line was supposedly of significance in the divergence of Yunnan flora.

Tanaka suggested a demarcation line between the citrus "*Archicitrus*" and "*Metacitrus*" groups, which extends in the northwest–southeast direction from Yunnan, China, to northern Vietnam [30]. This geographical line was given the name "Tanaka line of Citrus distribution" or the "Tanaka line" [30]. Later, this line was suggested to be the rough demarcation line between Sino-Himalaya and Sino-Japanese flora-based research on some genera of the Sino-Himalaya and Sino-Japanese distributions [56]. Combined with some orchid plants and their distributions, this line was renamed the "Tanaka–Kaiyong line" [57]. This line has been studied and discussed by many researchers [13,32,33,58–62]. However, Pang et al. studied the phylogenetic tree of *Citrus* based on an AFLP marker analysis, which did not support the "Tanaka line of Citrus distribution" [63]. We consider that the "Tanaka line" has no practical significance for the floristic divergence in Yunnan [64].

We also found geographic differences in the plants between the southwest, south, and southeast of Yunnan. Although dominated by tropical components and having the highest proportion of tropical Asian components, the flora of southeastern Yunnan has subtropical, temperate families with relatively abundant species numbers, such as Magnoliaceae, Theaceae, Cornaceae, Symplocaceae, Caprifoliaceae, and Aquifoliaceae. In particular, the characteristic families of East Asia and the Himalayas, such as Diapensiaceae, Dipentodontaceae, Eupteleaceae, Grossulariaceae, and Torricelliaceae, are present in the tropical flora of southeast Yunnan but absent in the flora of southern Yunnan [28]. Furthermore, 349 genera,

including 57 genera of East Asian distribution, 53 genera of northern temperate distribution, 22 genera endemic to China, and 17 genera of East Asian–Northern American disjuncted distribution, were found in southeast Yunnan but were absent in southern Yunnan. On the other hand, 237 genera in southern Yunnan were not found in southeast Yunnan. Although the southern and southeastern tropical regions of Yunnan have similar tropical monsoon climates and tropical rain forest vegetation, the floristic difference between them implicates that they possibly went through different evolutionary histories on their flora [24,34]. This is supported by the geological histories of these regions. Southeastern Yunnan was mainly derived from the South China geoblock, while southwestern and southern Yunnan were derived from the Shan-Thai geoblock [65,66] (Figure 7). However, a clear floristic divergence between the flora of southwestern Yunnan and southern Yunnan has not been found.

Based on physiognomic and floristic studies of the tropical and subtropical evergreen forests in Yunnan, two conspicuous ecotones were also found: at the 800–1200 m altitude, the tropical lowland seasonal rain forest changes to a tropical lower montane evergreen forest; at the 1800–2100 (2200) m altitude, the tropical lower montane evergreen forest changes to a subtropical (warm temperate) evergreen forest, almost completely replaced by subtropical (warm temperate) Himalayan and Chinese endemic tree flora [67].

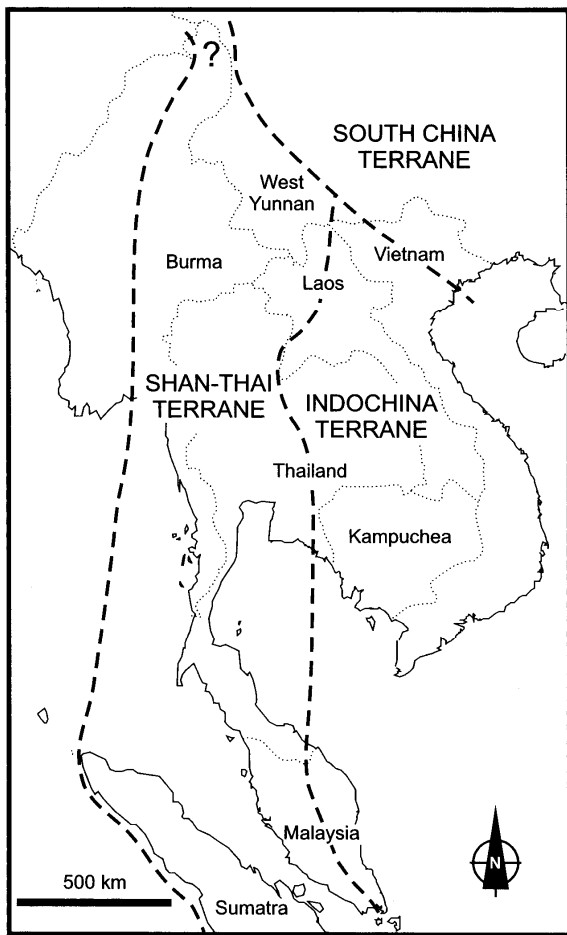

**Figure 7.** The main geoblocks in Southeast Asia (from Fortey et al., 1998).

## 3. Uplift in the Himalayan-Qinghai-Tibet Plateau and the Following Monsoon Climate Formation Affected the Evolution of Vegetation and Flora in Yunnan

The Indian plate moved northward and collided with the Eurasian plate in the early Cenozoic period (about 50 Ma (million years) ago), leading to the formation and uplift of the Himalayas [68]. With the Himalayan uplift, the Indochina geological plate was extruded toward Southeast Asia [69–73], ending in the late Miocene (10 Ma) [74]. Meanwhile, the Simao-Lanping geoblock in Yunnan moved toward the southeast with the Indochina plate

and underwent a clockwise rotation (about 30°) [75,76] and moved southeast by about 800 km, and its co-instantaneous clockwise rotation continued into the Miocene [71,77] (Figure 8). These geological events affected the origin and evolution of the flora and vegetation of Yunnan [11,27]. The Indian plate collided with the Asian plate, pushing away part of the Myanmar geological plate to the north by about 1000 km [78]. The northern movement of western Yunnan also occurred correspondingly, leading to the northwest–southeast inclined distribution pattern of tropical flora and tropical vegetation in Yunnan [17,26,79].

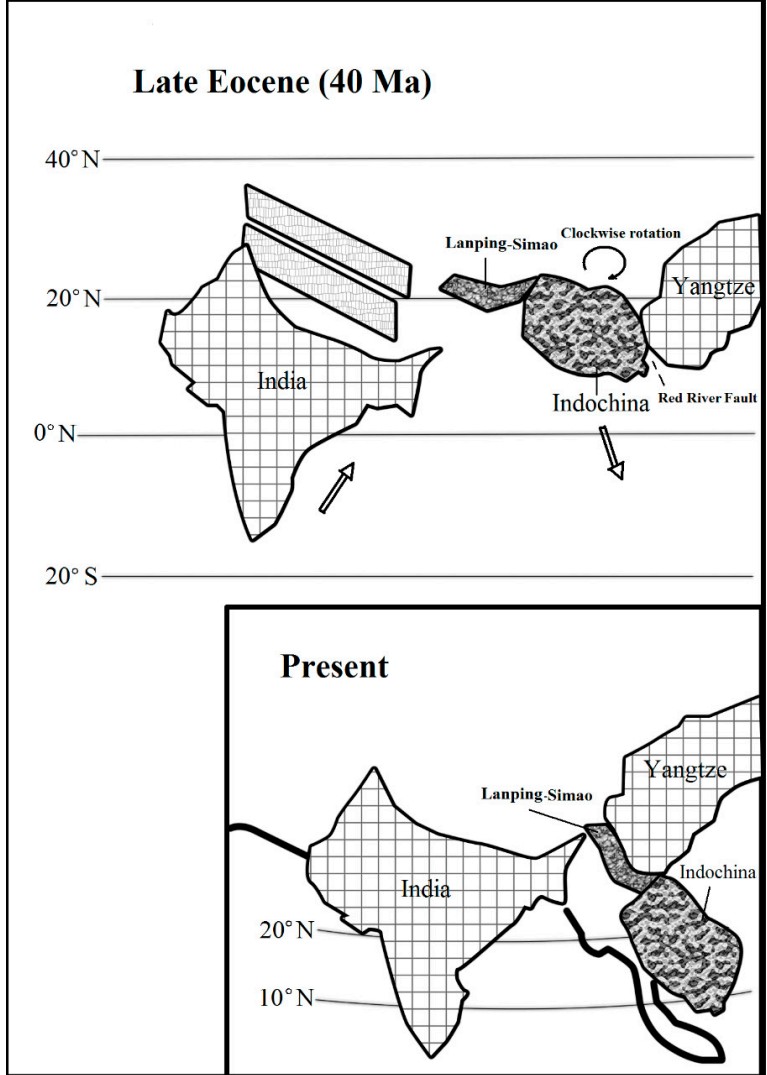

**Figure 8.** Clockwise rotation and southeastward extrusion of the Lanping-Simao and Indochina geoblocks during the late Eocene (redrawn from Sato et al., 2001, Figure 7).

The uplift in the Himalayan-Tibetan Plateau affected the global climate and widespread environmental changes since the late Cenozoic [80–82]. Due to the uplift of the Tibetan Plateau, the southwest monsoon occurring in South Asia emerged, which plays a decisive role in the development of tropical vegetation in India, mainland SE Asia and southwest China [83]. The formation and evolution of vegetation in Yunnan, especially tropical rain forests, are directly affected by the southwest monsoon, while the formation and evolution of Yunnan flora are obviously affected by historical geological events.

Much debate surrounds the timing of the monsoon climate in Southeast and East Asia. The mainstream view has been that, in the late Eocene, about 45–50 Ma ago, the Indian plate and the Eurasian plate collided and became integrated; however, the Himalayan-Tibetan

Plateau did not rise strongly but rather experienced a long process of uplift and deplanation at low altitudes (1000–2000 m alt.). The Himalayas were not very high until the Quaternary before 3.4 Ma or 2.5 Ma [81,84]. However, according to Su et al. and Liu et al., the Qinghai-Tibet Plateau should have risen earlier based on paleobotanical research [85,86]. When the Himalayas reached a considerable altitude (above 6000 m), the warm and humid air flow in the south was blocked by the high mountains, causing an abundance of precipitation on the southern slope, forming a warm and humid subtropical and tropical climate at lower altitudes and playing a decisive role in the development of tropical vegetation in SE Asia and southwest China [81,83]. Based on the simulation of late Oligocene paleogeographic data, Li et al. believe that the uplift of the northern Qinghai-Tibet Plateau during the Paleogene enhanced the East Asian monsoon climate system and drove the formation of humid and semi-humid vegetation types dominated by the evergreen broad-leaved forests in East Asia [87]. Undoubtedly, the formation of the Himalayas greatly influenced the strength of the southwest monsoon [88].

The geological and climatic histories of a region directly affect the formation and evolution of its flora and vegetation. The findings of paleobotanical research not only provide a basis for exploring the period when the Himalayan uplift occurred and the monsoon climate formed but also are a key factor in piecing together the evolutionary history of regional flora and vegetation. Jacques et al. revealed that ancient vegetation in central and south-central Yunnan during the Miocene were similar to the current subtropical forests [35], but in southern Fujian in southeastern China, during the Miocene, a large number of dipterocarp fossils were found [36,89], which show that the vegetation in southern Fujian was similar to those of the tropical rain forests of Southeast Asia [90]. The paleobotanical information for southwest and southeast China during the Miocene revealed different vegetation, meaning that southwestern and southeastern China should have different geological, climatic, and vegetational histories.

Our studies on the flora and vegetation in Yunnan suggest that their formation and evolution are closely related to the Himalayan uplift, the monsoon climate formation, and the various accompanying geological events [11,24,25,27,37]. The clockwise rotation and displacement of the Simao-Lanping geoblock to the southeast, and the extrusion of the Indochina geological plate toward Southeast Asia are supposedly the main geological events that directly affected the formation and evolution of flora in Yunnan, while the formation and strengthening of the southwest monsoon is a direct factor in the evolution of the evergreen broad-leaved forests and tropical rain forests found in Yunnan.

The modern rivers draining into the plateau at the border of SW China, i.e., the Jinshajiang, Nujiang, and Lancanjiang (the upper reaches of the Mekong River), were once tributaries into a single, southward flowing system—the paleo-Red River, which drained into the South China Sea since at least the late Miocene [23]. A disruption of the paleo-drainage was caused by river capture and reversal prior to or coeval with the initiation of the Miocene uplift of the Himalayas [23]. River captures interrupted these large rivers, which led to species isolation and differentiation, and further excited new taxa evolution in these river drainage areas.

## 4. Origin and Evolution of Yunnan Biodiversity

Plant diversity is closely related to the evolution of flora and vegetation. Yunnan flora and vegetation are considered to have evolved from the Tertiary tropical and subtropical flora and vegetation. With the uplift in the Himalayas and the extrusion of the Indochina geological plate into Southeast Asia, the cosmopolitan and northern temperate floristic components quickly speciated and proliferated in the high mountains in the north, evolving into the present-day temperate flora and vegetation, while in the southern lowland region, the tropical Asian floristic elements influenced and evolved into the current tropical flora and vegetation. However, the original ancient floristic components in the central region of Yunnan remained and were inherited [11,27].

Taking southern Yunnan as an example, tropical elements account for 78.3% of all genera, among which the proportion of tropical Asian distributions is the highest, accounting for 30.19%, showing that the flora of southern Yunnan is now tropical Asian flora in nature. Furthermore, regarding the floristic composition of the tropical rain forest in southern Yunnan, 80% of families, 94% of genera, and more than 90% of species are from a tropical distribution, of which about 38% of genera and 74% of species are from a tropical Asian distribution [7]. The large proportion of tropical Asian genera and species in the tropical rain forest in southern Yunnan, are supposedly closely related to the formation and strengthening of the southwest monsoon, and the extrusion of the Indochina geological plate into Southeast Asia. Obviously, the formation and strengthening of the southwest monsoon have led to the emergence of the tropical wet climate locally in southern Yunnan, which enabled tropical rain forests to emerge. According to paleobotanical data, from the late Cretaceous to early Tertiary in southern Yunnan, the representative vegetation was speculated to be dry subtropical or southern subtropical montane evergreen broad-leaved forests [91]; from the Eocene to the Oligocene, the main vegetation may have still shared characteristics from the previous period; from the Miocene to the Pliocene, the forest vegetation was speculated to mainly be an evergreen broad-leaved forest based on the southern subtropical or subtropical features in southern Yunnan [92–95]; and the tropical flora and rain forest in southern Yunnan likely evolved after the Pliocene [9,10].

Studies on the flora in northwestern Yunnan have revealed that the worldwide and northern temperate families and genera had proliferated and experienced rapid speciation during the Tertiary during the Himalayan uplift and under climate fluctuations, where they evolved into the current temperate flora [11,25]. Paleobotany and paleovegetation studies revealed that the accumulation of alpine plant diversity in Hengduan Mountain began during the early Oligocene [39]. Therefore, the flora and vegetation in the northwestern region of Yunnan may have begun in the early Oligocene.

Apart from the southern-most and the northern Yunnan, most of Yunnan, especially the central part originally covered by evergreen broad-leaved forests on plateau, lower and upper montanes, which were further divided into three major vegetation sub-types—monsoon evergreen broad-leaved forest on the lower montane in southern and southwestern Yunnan (with a southern subtropical climate); the semi-wet evergreen broad-leaved forest mainly on plateau (with a subtropical climate), both on limestone and acid soil habitats; and the mid-montane wet evergreen broad-leaved forest on the upper montanes (with a subtropical to temperate climate) [22]. At vegetation level, the three evergreen broad-leaved forest types diverged considerably in species composition, physiognomy, and biogeography, although they are commonly dominated by species of the families Fagaceae, Lauraceae, and Theaceae. The monsoon evergreen broad-leaved forest is extremely rich in species and is characterized by a tropical physiognomy. It is dominated by tropical Asian species, which is similar to the tropical lower montane evergreen forest in Southeast Asia. The semi-wet evergreen broad-leaved forest and the mid-montane wet evergreen broad-leaved forest were characterized by a subtropical physiognomy and are dominated by Sino-Himalayan and Chinese endemic species, which are unique in southwestern China [21]. All of the three forest types commonly have species-rich families, which tend to have cosmopolitan distributions, but the families with fewer species have other distribution types. At generic and specific levels, the semi-wet evergreen broad-leaved forest and the mid-montane wet evergreen broad-leaved forest showed similar biogeographical patterns in the proportions of tropical (45% and 44%, respectively) and temperate (46% and 48%) elements, with northern temperate distributions comprising the highest percentage (18% in the semi-wet evergreen broad-leaved forest and 20% in the mid-montane wet evergreen broad-leaved forest) of total genera, while in the monsoon evergreen broad-leaved forest in southern Yunnan, tropical elements comprised 79% of the total genera, with elements of tropical Asian distributions contributing the highest percentage (27%). The tropical species in the monsoon evergreen broad-leaved forest make up 70.52% of its total species, of which the tropical Asian distribution contribute 64.73% of the total species, while both of the semi-wet

evergreen broad-leaved forest and mid-montane wet evergreen broad-leaved forest have the majority of temperate distribution species, 72.88% on limestone and 78.67% on non-limestone habitats in the former, and 80.27% in the latter. Of their temperate distribution species, both of them have the Chinese endemic and Chinese-Himalayan species contributing to the majority (Zhu, 2021). The three vegetation subtypes of evergreen broad-leaved forest in Yunnan are commonly dominated by species of the families Fagaceae, Lauraceae, and Theaceae, which reveals their common early origin. Later their divergence took place, with events in the geological history of Yunnan caused by the uplift of the Himalayas.

Comparing the flora of northern, central, and southern Yunnan, they commonly include families of a tropical distribution, which make up the highest ratio, but in the genera and species, northern Yunnan is distinct from southern Yunnan [11]. This revealed that the Yunnan flora is identical to the tropical families that dominated before the uplift of the Himalayas. A floristic divergence took place with the uplift of Himalayas and with the formation of the monsoon climate. The northern region evolved into temperate flora and vegetation dominated by worldwide and northern temperate families and genera, while the southern region evolved with the extrusion of the geological plate to Southeast Asia, with tropical Asian element infiltration causing the evolution into tropical Asian flora [11,25]. East Asia is a relatively stable region in geological history, where the continuous evolution of flora and vegetation has occurred. Therefore, the flora and vegetation of central Yunnan are mainly from the East Asian flora and vegetation.

Deciduous forests with the same ecological physiognomy and structure as the ones in the Indo-Myanmar region appeared disjointedly in parts of the deep valleys of the Yuanjiang, Nujiang, Jinsha, and Lancang rivers and in some of the open valley basins most strongly affected by the monsoon. Regarding geographical elements, deciduous forests consist of 80% of the genera and 70% of the species that belong to tropical components, showing that they are tropical forests by nature. They are believed to have extensive associations with the tropical deciduous forests in the Indo-Myanmar region and occurred in a much larger area during some period during or before the Pliocene to Pleistocene, when a drier climate appeared in most parts of Yunnan. These forests have survived as refuges in hot dry valleys and basins possibly since the Pleistocene [18]. The higher rainfall in Yunnan during the later Pleistocene and early Holocene [96] likely explains their survival in these hot dry rain-shadow valleys. The tropical deciduous forest shrank to isolated dry habitats and valleys with expansion of evergreen broad-leaved forests with the increasing rainfall during the later Pleistocene.

Savanna-like vegetation is also commonly found in deep, hot and dry valleys within several large rivers strongly affected by the monsoon and by the foehn effect in Yunnan. Its flora is dominated by tropical families and genera and is fundamentally tropical in nature. An endemic genus *Tsaiodendron* (Euphorbiaceae), a newly published genus [97], is found locally in the savanna-like vegetation in the hot dry valley of Yuanjiang, SE Yunnan, and the compositae tree genus *Nouelia*, which is a mono-typical (*N. insignis*), is endemic to the hot dry valley of Jinshajiang, Northern Yunnan, offering evidence that these vegetations have existed in hot dry valleys for a long time. Some studies, such as those on *Musella lasiocarpa* (Musaceae) [98] and *Terminalia franchetii* (Combretaceae) [99,100], showed that their phylogenetic evolution could have been caused by the rapid uplift of Himalaya and concomitant river captures.

The geological events that have occurred since the Cenozoic, especially the river capture caused by the rise of the Himalayas and the climatic fluctuation, have influenced the evolution and divergence of the flora of the savanna-like vegetation in Yunnan [17]. The flora in hot dry valleys of the Yuanjiang, Nujiang, and Jinshajiang were compared. At the family level, the flora from each were almost the same, with a similarity coefficient of more than 87% (Similarity coefficient between A and B = the number of taxa shared by both A and B divided by the lowest number of taxa of A or B, multiplied by 100%). At the generic level, the highest similarity was between the Yuanjiang and Jinshajiang (73.84%), and all have a similarity of more than 62%, showing that close floristic affinity remains. However,

at the species level, the similarity coefficients were generally low, with the highest (53.76%) being between the Yuanjiang and Jinshajiang and the lowest (35.03%) being between the Yuanjiang and Nujiang. The flora of the hot dry valleys in Yunnan has diverged at the specific level [17].

The current flora in Yunnan can be categorized into five historical phytogeographical elements: East Asian subtropical floristic element (mainly in central Yunnan), Southeast Asian tropical floristic element (mainly in the southwest, south and southeast of Yunnan), the Himalayan floristic element (mainly in the northern region), the Mediterranean residual flora (widespread in Yunnan), and the Gondwana flora (mainly in the dry hot river valleys). Morley suggested that the Indian plate drifted to low latitudes in Asia during the Eocene, bringing the floristic components of Gondwana to Southeast Asia, which successfully evolved in Southeast Asia after 41 Ma [101]. Later, these Gondwana floristic elements also entered the dry and hot river valley regions of Yunnan province [17]. In paleogeography, Yunnan was located on the southeast edge of Eurasia. After the Indian plate and Asian plate collided, the Neo-Tethys closed and the Himalayan Plateau was uplifted. The sclerophyllous and microphyllous shrubs and maquis-like vegetation with Mediterranean floristic affinity in warm dry valleys in northern Yunnan remained, and the wide-spread sclerophyllous oak forests in Yunnan evolved from the Mediterranean-related flora [102,103].

In summary, the formation and evolution of plant diversity in Yunnan are closely related to the geological events and the formation and strengthening of the monsoon climate.

## 5. Conclusions

In the eyes of a biologist, the flora of Yunnan of SW China make up a mysterious kingdom. Yunnan only takes up 4.1% of the total land area but includes more than half of all plant species found in China and has various types of vegetation, including tropical rainforests to cold temperate coniferous forests. The complex topography, landforms, and huge difference in altitudes have created diverse habitats in Yunnan. In physical geography and biogeography, Yunnan is a transitional region located in the tropical SE Asia and subtropical-temperate East Asia and Himalaya. Located at the particular intersection between SE Asia, East Asia, and Himalaya, the possibilities for its biota origins are plentiful, because SE Asia, East Asia, and Himalaya offer it ample biodiversity sources. In geological history, it has experienced geological events such as the extrusion of the Indochina geoblock, the fracture and dislocation of fragments of smaller geoblocks, and large rivers' captures due to the uplift in the Himalayas. It also experienced the formation of a monsoon climate due to the Himalayan uplift. These particular natural and historical conditions serve as the basis for the evolution of such high biodiversity in Yunnan.

In this unique geographical area, these geological events, monsoon climate formation and strengthening, and climate fluctuation evidently led to frequent species migration, isolation, and divergence, and to quicker-than-usual speciation. Based on paleobotanical studies, the majority of the vegetation and flora in Yunnan may have formed during the period from the Oligocene to the Miocene. With a further uplift in the Himalayas, a topographically differential uplift, the strengthening of monsoon climates, and climate fluctuation after the Pliocene, the south–north vegetation and the flora differences increased, and west–east biogeographical divergence became clear. Abundant biodiversity has arisen and evolved accordingly in Yunnan.

**Author Contributions:** Conceptualization, H.Z.; Methodology, H.Z. & Y.T.; Software, H.Z. & Y.T.; Validation, H.Z. & Y.T.; Formal analysis, H.Z. & Y.T.; Investigation, H.Z. & Y.T.; Resources, H.Z. & Y.T.; Data curation, H.Z. & Y.T.; Writing—original draft preparation, H.Z. & Y.T.; Writing—review and editing, H.Z. & Y.T.; Visualization, H.Z. & Y.T.; supervision, H.Z.; Project administration, H.Z. & Y.T.; Funding acquisition, H.Z. & Y.T. All authors have read and agreed to the published version of the manuscript.

**Funding:** This project was funded by The National Natural Science Foundation of China (41471051, 31970223).

**Institutional Review Board Statement:** Not applicable.

**Data Availability Statement:** Not applicable.

**Conflicts of Interest:** The authors declare no conflict of interest.

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
