# Peer review of "Flora and Vegetation of Yunnan, Southwestern China: Diversity, Origin and Evolution"

_diversity, doi:10.3390/d14050340_

Round 1

Reviewer 1 Report

Is Figure 4D really sclerophyllous oak forest in northern Yunnan?  The photo makes it appear to be more like a naturalized Eucalyptus forest.  

The 'Flora and vegetation of Yunnan, southwestern China: diversity, origin and evolution' is a good summary of the geological events and vegetation history and vegetation types of Yunnan Province, China.  It is slightly more focused on the vegetation of southern Yunnan, then on the flora and vegetation of the hot dry valleys and then the high elevation vegetation in the north, particularly in the northwest, but says little about the vegetation of central and northeastern Yunnan.  Perhaps that is due to the lack of good, natural vegetation in those areas, which have been greatly modified by the activities of humans.  What about the vegetation in the central part and the part of the province around greater Kunming and in the karst regions?

Author Response

Is Figure 4D really sclerophyllous oak forest in northern Yunnan?  The photo makes it appear to be more like a naturalized Eucalyptus forest.  

Answers: This photo is a sclerophyllous oak forest dominated by species Quercus guajavifolia and Q. semecarpifolia.

The 'Flora and vegetation of Yunnan, southwestern China: diversity, origin and evolution' is a good summary of the geological events and vegetation history and vegetation types of Yunnan Province, China.  It is slightly more focused on the vegetation of southern Yunnan, then on the flora and vegetation of the hot dry valleys and then the high elevation vegetation in the north, particularly in the northwest, but says little about the vegetation of central and northeastern Yunnan.  Perhaps that is due to the lack of good, natural vegetation in those areas, which have been greatly modified by the activities of humans.  What about the vegetation in the central part and the part of the province around greater Kunming and in the karst regions?

Answers: We added a paragraph to clarify the dominant evergreen broad-leaved forests on central plateau, lower montanes in south and upper montanes in central and central north in Yunnan, including both on limestone and acid soil habitats. In this paragraph, the floristic composition and biogeographical elements are concisely enumerated. Its origin and divergence are discussed.

Reviewer 2 Report

The subject of the paper is very interesting. I appreciated reading it. Surely, a great wealth of accumulated knowledge is evident in the discourse of the author. It is a knowledgeable person.  Nevertheless, I do not really understand what this text is meant to be. It looks like an overview one would expect to find in a textbook or as the transcription of an oral conference, surely not a scientific paper.  It is not organized as a scientific paper: objectives- or scientific questions, methods, data sources are not clearly identified or not used. I would expect a phylogeographical meta-analysis of published data, for instance.  The biogeographical analysis is too vague and based alone on the comparison of plant percentages from - also vague or ill-defined - biogeographical (or bioclimatic) regions. The text, in general, lacks formal precision. E.g. biogeographical system used. The explanation of vegetation types is outdated and is almost merely a re-exposition of a previous system with little knowledge being added to it.  Moreover, 40% of references are of the own author. This would be acceptable in a summary for a textbook only. Not in a scientific article.   Finally, the text has conceptual imprecisions and sometimes tautologies; statements that are vague, meaningless, speculative and unsupported either by data or references.  These are too many to pinpoint specifically.   It has a flawed English language, too.

My opinion is that the text is not suited for a scientific paper, accounting for the quality standards in the Diversity journal and I would recommend rejection.

Best regards.

Author Response

Answers: First, thank the reviewer’s sincere comments to the manuscript. This is a review paper, not a standard research article. On my understanding to a review article, it is a comment article, in which it will summarize the current state of the referred affairs, and will give a personal review on the aspects of the affairs. In Yunnan province of southwestern China, the floristic and vegetational studies at provincial and regional levels are a fewer. Most of additional floristic and vegetational studies focused on nature reserves or small regions. We worked on the floristic and vegetational studies in Yunnan for more than 30 years, therefore a lot of the references in the review article is own authors, which are inevitably the basis of the flora and vegetation studies in the review article. More other references of flora and vegetation studies at provincial and regional levels could not be retrieved. The aim of this article is to make an overall review in the flora and vegetation of Yunnan in a special issue in the Diversity Journal asked by the guest editor of the special issue. If I misconceive in some points of view in the review article, I hope that the reviewer could give understanding.

Reviewer 3 Report

Flora and vegetation of Yunnan, southwestern China: diversity, origin, and evolution

The paper investigates flora and vegetation of Yunnan from a biogeographical point of view. Composition and characteristic of the flora and vegetation have been related with the geological and climatic histories of the region in order to trace their origin and evolution.

The paper is well written. The aim of the paper is explained in the introduction section: to carry out an overview of the high plant biodiversity of the whole region.

It’s a Review, so methodology is not requested.

I think the paper could be accepted in the preset form, but only a few observations:

pg 10, line 290: Torricelliaceae instead of Toricelliaceae

pg 15 lines 40-486: I suggest a very short indication on how the evaluation of similiarity/similarity coefficient has been assessed

Author Response

pg 10, line 290: Torricelliaceae instead of Toricelliaceae

Done!

pg 15 lines 40-486: I suggest a very short indication on how the evaluation of similiarity/similarity coefficient has been assessed

Done!

Round 2

Reviewer 2 Report

Dear authors

My objections were based on the assumption that the article was an overview or synthesis. I repeat: I enjoyed reading it. Therefore, although of evident scientific quality, its form was not of a scientific article in the current sense. Nevertheless, after acknowledging the author´s explanations, especially that the text issues from an explicit invitation from editors to produce a review article, I should change my mind. Moreover, I do not see any relevant flaws worth mentioning and I agree with the comments made by reviewers #1 and #3. 
Therefore, I should change my recommendation to editors, to accept the article. 

Best wishes.

A reviwer.